# Synthesis of sEMG Signals for Hand Gestures Using a 1DDCGAN

**DOI:** 10.3390/bioengineering10121353

**Published:** 2023-11-25

**Authors:** Mohamed Amin Gouda, Wang Hong, Daqi Jiang, Naishi Feng, Bin Zhou, Ziyang Li

**Affiliations:** Department of Mechanical Engineering and Automation, Northeastern University, Shenyang 110819, China; mohamedamin@stumail.neu.edu.cn (M.A.G.); 1910103@stu.neu.edu.cn (D.J.); fengnaishi98@163.com (N.F.); 1810097@stu.neu.edu.cn (B.Z.); 2110108@stu.neu.edu.cn (Z.L.)

**Keywords:** DCGAN, EMG, sEMG, AI, bio-signals, data augmentation, classification

## Abstract

The emergence of modern prosthetics controlled by bio-signals has been facilitated by AI and microchip technology innovations. AI algorithms are trained using sEMG produced by muscles during contractions. The data acquisition procedure may result in discomfort and fatigue, particularly for amputees. Furthermore, prosthetic companies restrict sEMG signal exchange, limiting data-driven research and reproducibility. GANs present a viable solution to the aforementioned concerns. GANs can generate high-quality sEMG, which can be utilised for data augmentation, decrease the training time required by prosthetic users, enhance classification accuracy and ensure research reproducibility. This research proposes the utilisation of a one-dimensional deep convolutional GAN (1DDCGAN) to generate the sEMG of hand gestures. This approach involves the incorporation of dynamic time wrapping, fast Fourier transform and wavelets as discriminator inputs. Two datasets were utilised to validate the methodology, where five windows and increments were utilised to extract features to evaluate the synthesised sEMG quality. In addition to the traditional classification and augmentation metrics, two novel metrics—the Mantel test and the classifier two-sample test—were used for evaluation. The 1DDCGAN preserved the inter-feature correlations and generated high-quality signals, which resembled the original data. Additionally, the classification accuracy improved by an average of 1.21–5%.

## 1. Introduction

Muscle contractions create electromyography (EMG) signals, which can be used to measure muscle activity and to diagnose muscle disorders. These EMG signals can be captured using surface, needle and intramuscular electrodes [1,2,3]. However, the prevalent approach in research and prosthetic applications involves the utilisation of surface electromyography (sEMG) due to its ability to capture global muscle information, in contrast to needles, which only gather data from a single muscle. Moreover, the utilisation of needles and intramuscular electrodes presents potential drawbacks, such as pain and inherent risks.

Various privacy laws may impose restrictions on researchers who require access to sEMG signals from specialised companies. Researchers may find it challenging to carry out repeatable research and product development as a result, which limits replicable research and product development. Researchers created multiple datasets, including those mentioned in Refs [4,5] and others, to provide independent data sources. Nevertheless, these datasets might have missing or damaged data [6] or lack the gesture repetitions needed to build a robust classification model, resulting in overfitting, according to Kaczmarek et al. [7]. Some researchers represented the sEMG signal in two dimensions and employed translation and rotation to augment the data, but these methods are not appropriate for one-dimensional time series data [8].

In order to overcome these issues, generative adversarial networks (GAN) can be utilised to generate new data instances. There are various methods to generate data. For example, general-purpose parametric models can also be used to create data in the easiest way [9]; however, they are the least accurate, and these signals can be immediately identified as fabricated data. Expert-driven parametric models are more accurate but require more experience. Anonymised bio-signals can be used to deliver data in a private manner. For both the control and diseased groups, it is imperative to adhere to experimental best practices when recording, processing and storing signals. Data re-identification is a possibility if appropriate collecting procedures are not followed. Deep-learning models such as GAN have been used to synthesise bio-signals by learning the statistical patterns of the original data and producing synthetic data. GANs demand deep-learning expertise but little domain knowledge.

GANs can let organisations share synthesised signals with researchers without breaking patients’ data privacy laws, boosting research repeatability. All approaches must protect patient privacy and follow privacy standards. The synthesised data have the potential to be utilised for the purpose of training AI-controlled prostheses, thereby enabling the achievement of precise control. Typically, in the context of an amputee using an AI-controlled prosthetic device, sEMG electrodes are attached to the individual’s muscles. Subsequently, the amputee proceeds to execute various hand gestures, with this iterative process being repeated multiple times until the AI model achieves convergence. The act of iteratively performing the gesture on multiple occasions is exhausting and introduces complexity to the overall procedure. GANs have the potential to expedite the convergence of the classification model by generating synthetic data. Furthermore, GANs may potentially avoid the necessity for amputees to visit a laboratory, allowing them to perform the process in the comfort of their own homes. Data synthesis protects patient privacy and enhances research.

### Related Work

GANs were initially proposed by Ian Goodfellow; it is a type of machine-learning (ML) model, which can be used to generate realistic data. GANs work by pitting two neural networks against each other: a generator, which creates new data, and a discriminator, which tries to distinguish between real and fake data [10].

Research on GANs has grown significantly, with early work focusing on generating images [10,11]. Later, researchers began to focus on generating text [12] and speech [13]. More recently, researchers have begun to consider using GANs to generate data, which are not easily accessible, such as medical images or financial data. The generation of medical data has become increasingly important with the advancement in pattern recognition techniques and the rise in privacy concerns. Generating medical data focuses on finding new deep-learning models, which can generate quality data and ways to evaluate them. Esteban et al. proposed RGAN and RCGAN time series data models [14]. Visual and quantitative methods, such as sample likelihood and maximum mean discrepancy (MMD), were used to examine the data. Train on synthetic, test on real (TSTR) and train on real, test on synthetic (TRTS) are novel evaluation methods, which use the generated data to train a classifier and the actual data to evaluate it and vice versa. The authors designed three tests—one qualitative and two statistical—to mitigate overfitting due to privacy concerns regarding sensitive material.

Hazra et al. [15] developed a one-dimensional GAN called SynSigGan, which works well on biological signals. Their model refines signals using discrete wavelet transform, thresholding and inverse discrete wavelet transform and then evaluates the generated data using the Pearson correlation coefficient, root mean square (RMS) error, per cent RMS difference, mean absolute error and Fréchet distance (FD) for statistical analysis. Researchers suggested a unique method employing bidirectional RNN and statistical stages to generate synthetic biological signals for patients or events. The method generates five biological signals and outperforms other generative models in evaluation metrics.

Beaulieu-Jones et al. [16] noted that sharing individual-level clinical study data is hindered by challenging requirements. The sharing of data among academics is often hindered by the need for formal collaborations and elaborate data usage agreements. Deep neural networks were combined to create a systolic blood pressure trial (SPRINT). Differential privacy, which reduces the likelihood of identifying clinical research participants, was used to train neural networks. Hypothesis-generating analyses simulated the original trial data using machine-learning predictors from the synthetic population. The discovery suggests that synthetic data enable secondary studies and replicable exploration of clinical datasets while protecting participant confidentiality.

Anicet Zanini et al. [17], in an effort to solve the lack of available Parkinson’s data, proposed approaches based on DCGAN and a combination of DCGAN and style, introducing two new data augmentation methods using DCGANs and style transfer for Parkinson’s disease sEMG signals. The proposed models can simulate individual patients’ tremor patterns and extend them to different movement protocols. This allows for the extension of patient datasets and the generation of tremor simulations for validating treatment approaches in different movement scenarios. The experimental results demonstrate the models’ ability to adapt to various frequencies and amplitudes of tremors, making them a promising tool for Parkinson’s disease research. Moreover, fast Fourier transform (FFT), mean square error, dynamic time wrapping (DTW) and sEMG envelope cross-correlation are the metrics used in their research to evaluate the generated signals. The mean square error (MSE) of FFT magnitudes is used to measure the similarity between time series signals. DTW is used in time series analysis to measure the similarity between sequences by comparing local cost functions. Cross-correlation measures the similarity between two series by considering their displacement.

Campbell et al. [18] developed a framework for brief training programmes using subject-specific synthetic sEMG data using SinGAN to generate synthetic data. When combined with restricted training, artificial data can improve classification accuracy. The study created 1000 fake sEMG segments comprising six actions. Qualitative, quantitative and classification methods examined the results. Artificially created data improved classification accuracy by 5.4% in cross-validation testing. This method could improve myoelectric control training in data-poor situations.

Zhang et al. [8] used one-dimensional energy-based generative adversarial networks (EBGAN) to generate sEMG features to improve classification precision. The discriminant network uses the energy paradigm instead of binary assessment, and the fully linked layers capture the distribution of genuine sEMG data to create comparable data. This GAN design achieves a lower MMD than others. This data augmentation method increased traditional classification model precision by 1.21–5%.

In brief, numerous studies have been conducted in the field of GANs, with a particular emphasis on their application in image generation. Subsequently, GANs have been employed for the analysis of various types of data, including bio-signals. Many studies approach time series bio-signals as two-dimensional images, which may not be suitable for this particular type of data. Furthermore, it is crucial to contemplate the establishment of appropriate metrics for the said signals.

To this end, this research employs a 1DDCGAN with DTW, FFT and wavelets as inputs to the discriminator to generate the synthetic sEMG of hand gestures. This method generates synthetic data to expand the datasets. Two different sEMG datasets were used in this research to validate this approach. In order to evaluate the quality of data generated by the model, several evaluation techniques were employed, including the classifier two-sample (C2ST) test, the Mantel test and conventional classification metrics, such as classification accuracy and augmentation test. The proposed model improved the classification accuracy by 1.21–5% in the augmentation test. The two novel metrics showed that the model produced high-quality data, which resembled the original data and kept feature interrelationships. To the author’s knowledge, the usage of 1DDCGAN, the C2ST and the Mantel test is new in the context of EMG hand gestures.

This research paper is divided as follows:Section 1: Introduction, which provides the background on sEMG signals and the challenges with acquiring data, as well as an overview of using GANs to generate synthetic medical data.Section 2: Materials and Methods, which details the data acquisition, signal processing, proposed 1DDCGAN architecture and evaluation methods, including the Mantel test, classification, augmentation test and classifier two-sample test.Section 3: Results, which presents the results of the Mantel test, classification, augmentation test and classifier two-sample test in assessing the quality of the synthesised sEMG signals.Section 4: Discussion, which analyses and interprets the results, comparing them with previous literature.Section 5: Conclusion, which summarises the efficacy of using 1DDCGAN to generate synthetic sEMG signals based on the evaluation metrics.

## 2. Materials and Methods

The objective of this study is to create a GAN, which can produce high-quality signals capable of reproducing the characteristics of the original hand gesture signal, as well as augmenting the datasets, which lack repetitions or missing data points to improve the classification. Figure 1 illustrates the proposed model’s flow chart. This section will provide a comprehensive overview of the techniques employed in conjunction with our proposed strategy.

Figure 1 depicts the flow chart representing the methodology employed in this study. The sEMG signals obtained from the respective hand gestures and the synthetic signals generated by the generator are initially subjected to noise filtration. Subsequently, the DTW, wavelets and FFT are extracted from both signals. These extracted features, along with the original signals, are subsequently input into the discriminator. Finally, after the training of the 1DDCGAN, the generated signals and the filtered original signals are evaluated using the augmentation classification method to measure the improvement the generated signals provide, classification to examine the accuracy of the generated and original dataset independently, C2ST to examine the ability of pattern recognition to distinguish between the two datasets and the Mantel test to examine whether the relationship between features is preserved. The data acquisition setup and the devices used are detailed in Section 2.1. Signal filtering and the extracted features are explained in Section 2.2. The proposed architecture and the GAN network theory are detailed in Section 2.3. Finally, the evaluation methods employed are explained in Section 2.4.

### 2.1. Data Acquisition

In this study, two distinct datasets were used to assess our approach’s efficacy in synthesising various sEMG classes. The first dataset was private, and the second was public [19]. Both datasets were collected with two electrodes, whereas the current trend uses an array [4,5,7,20,21]. The usage of fewer electrodes is better in some applications because it can help reduce hardware and computation costs, which can be an essential consideration in practical settings.

For the private dataset, five non-disabled participants were instructed to perform six different gestures. They were two male and three female volunteers between the ages of 26 and 29 and were right-handed. The participants sat with their arms extended on a disc during the recording. Participants were briefed before the experiment. A video showed participants when to hold and release movements during the session. In order to prevent overfitting and reduce variation, the participants were asked to perform different gestures in a sequence rather than repeating the same gesture multiple times consecutively [7]. Each session included six motions, held for three seconds, followed by three seconds of rest; this procedure was repeated 15 times over three days to avoid muscle fatigue. Figure 2 shows the six gestures, the electrode placement and the timeline of the experiment. The apparatus used to collect the private dataset was Myon Atkos-mini. It contained two wireless sEMG sensors with a 2000 Hz sampling frequency. The electrodes were placed on the flexor and extensor digitorum muscles. The experiment was conducted in the Mechatronics laboratory at Northeastern University in Liaoning, China. The procedures of the experiment were in accordance with the university’s ethical and other relevant committees, and the participants also provided written consent.

The publicly available dataset used in this research was the dataset from Ref [19]. Six men and two women between 20 and 35 years old participated. The subjects were asked to perform ten different gestures, each held for five seconds, then repeated six times. The sEMG data were collected using two channels of the Delsys DE 2 series with a frequency of 4000 Hz. The sensors were placed on the biceps brachii and triceps brachii muscles.

### 2.2. Signal Processing

Signal processing techniques are applied to EMG data for multiple purposes, including noise reduction, artefact removal and feature extraction.

#### 2.2.1. Filtering

Before the signal was used in the GAN network, it was filtered, so that the generated signal did not have artefacts and could be easily processed by researchers. A bandpass Butterworth filter of the fourth order with pass range set to 10–500 Hz, as it is known to encompass the frequencies associated with muscle activity [21]. Additionally, a notch filter was tuned to 50 Hz, which is the frequency at which power line interference occurs.

#### 2.2.2. Feature Extraction

After the GAN signal generation and before the evaluation stage, features were extracted using the windowing technique, which divides the signal into smaller, overlapping segments called windows. Windowing is particularly useful when the signal being analysed is non-stationary. Sliding windows of various lengths, notably 200, 400, 600, 800 and 1000 data points, were used in the analysis. The amount of overlap was set at 25%, 50%, 75% and 100% of the window length. Afterwards, features were extracted from these different windows with different overlaps.

The extracted features consist of a collection of features in both the time domain and time-frequency domain [22,23,24,25,26] (Equations (1)–(7)). The time domain features used consist of the mean absolute value (MAV_w_), slope sign change (SSC), wavelength (WL), mean absolute value slope (MAVS), histogram (HIST), zero crossing (ZC) and RMS. The marginal discrete wavelet transform (mDWT) was selected as the time-frequency domain feature. SSC is a time domain feature, which represents the number of times the sEMG signal slope changes signs in a specific time window. ZC is a time domain feature, which refers to the number of times the sEMG signal crosses the zero-axis in a time interval, which reflects the frequency of signal values changing signs inside a window. MAV_w_ is a time domain feature, which is a statistical measure of the absolute magnitude of the sEMG signal.
(1)$$MAV_{w}(x)=\frac{1}{T}\sum \left|x_{t}\right|$$WL is a time domain feature, which is a measure of the total length of the sEMG signal over a given window of time and is indicative of the complexity of the signal.
(2)$$WL_{w}(x)=\sum_{t=2}^{T}\;\left|x_{t}-x_{t-1}\right|$$MAVS is a time domain feature, which refers to the rate of change in the MAV between two adjacent windows.
(3)$$MAVS_{w}(x)=MAV_{w+1}(x)-MAV_{w}(x)$$The integrated absolute value (IAV) is a time domain feature, which measures the area under the absolute value of the sEMG signal curve within a given window of time.
(4)$$IAV_{w}(x)=\sum_{t=1}^{T}\;\left|x_{t}\right|$$HIST is a time domain feature, which measures the sEMG signal amplitude distribution throughout time. The number of sEMG signal values in each bin is counted to determine HIST.
(5)$$HIST(x)=hist\left(x_{1:T},B\right)$$RMS is a measure of the amplitude of the sEMG signal over a certain time.
(6)$$RMS_{w}(x)=\sqrt{\frac{1}{T}\sum_{t=1}^{T}\;x_{t}^{2}}$$
mDWT is a time-frequency domain feature, which measures the signal’s frequency content at different scales and is obtained through a wavelet transform. The wavelet transform was created with three levels, and Daubechies 7 wavelet, *l*, is the deepest level of the decomposition.
(7)$$\begin{array}{c} mDWT_{l}(x)=\sum_{\tau =0}^{T/2^{l}-1}\;\left|\sum_{t=1}^{T}\;x_{t}\psi_{l,\tau }(t)\right|\\ \psi_{l,\tau }(t)=2^{-\frac{m}{2}}\psi \left(2^{-l}t-\tau \right) \end{array}$$

### 2.3. Generative Adversarial Network

GANs are effective machine-learning generative models, which are able to generate realistic objects, which are hard to distinguish from realistic ones. A GAN is composed of two distinct neural networks: the generator and the discriminator. The network’s task is to learn the underlying distribution of a dataset and use that knowledge to generate new data similar to the original data.

For a layered perceptron, the generator is modelled as a differential function G, which maps the latent space ζ as G(ζ; θg), while θg are the perceptron parameters. In terms of the discriminator, it is modelled as D(x; θd) with a scalar output, and x is the real input data [10]. The discriminator (D) and the generator (G) are trained concurrently. The discriminator’s task is to maximise the probability of correctly assigning labels to both the synthetic signal G(ζ) and the realistic signal x, while the generator’s task is to minimise the log (1 − D(G(ζ))). This is the minimisation–maximisation of the value function V (G, D) [10]:(8)$$\underset{G}{min}\;\underset{D}{max}\;V(D,G)=E_{x∼p_{\mathrm{data}\;}\left(x\right)}[log\;D(x)]+E_{z∼p_{\zeta }\left(\zeta \right)}[log\;(1-D(G(\zeta )))]$$

Figure 3 demonstrates the alternate training of the generator and the discriminator. The discriminator receives synthetic examples $\hat{X}$ produced by the generator from input noise and real examples (X). Afterwards, the discriminator makes predictions $\hat{Y}$, which is the probability score of how synthetic and how real each of the signals is. The predictions are compared using BCE loss with the desired labels for synthetic and real signals. This process leads to the update of θ_d_, where d denotes the parameters for the discriminator. The generator is only updated by θ_g_ (g denotes the generator), which are the parameters resulting from the synthetic signals only. That is, the generator training is not affected by the real signals.

After cost computation, the gradient is propagated backwards, and the parameters of the generator θg are updated. As the alternate training occurs, models are trained in succession. Therefore, in the alternate training of GANs, it is important for both models to improve concurrently, and they should be kept at similar skill levels from the beginning of training. In terms of the GAN loss, this research used the BCE loss function (Equation (9)). The BCE loss measures the divergence between two probabilities; in the context of a GAN, the predicted probability distribution is the output of the discriminator, which indicates whether the input is real (1) or synthetic (0).
(9)$$J\left(\theta \right)=-\frac{1}{m}\sum_{i=0}^{m}\underset{L1}{\underbrace{{[y}^{\left(i\right)}\log\;h\left(x^{\left(i\right)},\theta \right)}}+\underset{L2}{\underbrace{{(1-y}^{\left(i\right)})\log\left(1-h\left(x^{\left(i\right)},\theta \right))\right]}}$$

h: prediction,X: features,θ: parameters,y: label,J: average patch loss.

In Figure 4, the L1 part of the equation approaches zero when the real value is correctly detected, while the L2 component of the equation yields zero when a synthetic signal is successfully identified. If the model incorrectly classifies either part’s relevant label, a significant loss will result for both.

#### Proposed Architecture

The 1DDCGAN network was implemented in Python, while MATLAB was used to extract features from the sEMG signals to be classified during the evaluation phase. Latent noise is fed into the generator, which produces a synthetic sEMG signal. FFT, DTW and wavelets are extracted from the synthetic and real signals and then fed into the discriminator, which distinguishes between real and synthetic signals and updates the generator accordingly. The discriminator in this study used the DTW, FFT and wavelet techniques, which were adopted from previous work [19] but in a different context, which employed GANs and style transfer to produce sEMG signals mimicking the tremor pattern of individuals with Parkinson’s disease. The present study did not incorporate the style transfer component.

The layers of the generator and the discriminator both used batch normalisation, dropout, 1D convolution and leaky ReLU. Batch normalisation ensures that the values input to each layer have constant scales and distributions across the batch, reduces internal covariate shifts and enhances training stability and convergence, using mini-batch statistics to normalise the data input to each layer [27]. Dropout is a regularisation approach, which is intended to enhance generalisation and lessen overfitting. The generator and discriminator networks use 1D GANs and convolutional layers to learn and extract meaningful features from the input data. Negative values can have a non-zero slope using the leaky ReLU activation function, preventing mode collapse.

Generator

The generator network consists of six layers: convolution, batch normalisation, dropout and ReLU activation. Three of them have an up-sampling layer to increase the input dimensions. The six layers are connected to a dense layer and a Tanh activation at the end (Figure 5).

In the layers of the generator, up-sampling and convolution are used instead of deconvolution because deconvolution can be computationally expensive and introduce artefacts into the signal. Up-sampling is used to increase the size of the input by creating new data points between existing data points; in this paper, nearest neighbour interpolation was used.

The output of the generator contains a hyperbolic tangent (tanh) activation function—which prevents the generator from producing unrealistic or out-of-range samples—and increases GAN training stability and convergence, resulting in higher quality output.

Discriminator

The input to the discriminator is the generator’s output and the participants’ original signals. The input is passed through the convolution layer, and the output is concatenated with the output of the same convolution layer but with input from the mini-batch discriminator, FFT, DTW and wavelet filters. In Figure 6, the discriminator consists of seven layers containing convolution, batch normalisation, leaky ReLU and dropout.

Salimans et al. (2016) [27] proposed several recommendations to improve GAN performance, including one-sided label smoothing to improve training stability and prevent mode collapse. This works by substituting hard labels (0/1) with soft labels (0.1/0.9) to train the discriminator.

By replacing the target labels with real examples with a slightly lower value (denoted as α) and synthetic signal targets (β), the discriminator model can be described as
(10)$$D(x)=\frac{\alpha p_{\mathrm{data}\;}(x)+\beta p_{\mathrm{model}\;}(x)}{p_{\mathrm{data}\;}(x)+p_{\mathrm{model}\;}(x)}$$When the $p_{\mathrm{model}\;}$ is large and the $p_{\mathrm{data}\;}$ are very small in value, the samples have no incentive to move closer to the data. Therefore, the positive labels (α) are only smoothed.

### 2.4. Evaluation

It is difficult to judge the quality of the images generated by GANs, let alone of those generated by sEMG signals. When it comes to images, humans can be employed to assess their quality. However, it is not feasible to have humans judge the quality of synthetic sEMG signals. This presents an obstacle to assessing the quality of generated sEMG signals and requires alternative methods, such as objective metrics or expert evaluation, to evaluate the effectiveness of the GAN. The features from Section 2.2.2 will be used for evaluation, since they offer more insightful data regarding the current window than the raw signal.

Salimans et al. [27] introduced the inception score (IS) and the Fréchet inception distance (FID). The IS is an objective metric for evaluating the generated images, while the FID is used to measure the distance between the features of the synthetic dataset and the realistic one.

Delaney et al. [28] introduced DTW as a metric for time series data, and it has been used successfully in Ref [17] to evaluate the model’s performance. Esteban et al. [14] quantitively used sample likelihood and MMD to compare the synthetic ICU data with the original dataset. Hazra et al. [15] and Hernandez-Matamoros [29] used the RMS error, percentage RMS and other metrics to evaluate the generated dataset.

This study utilises the Mantel test and C2ST and classification techniques to evaluate the quality of the generated signals following the training process. Furthermore, the ML models were utilised to classify both the original signal and the signals generated through synthetic signal augmentation to investigate the impact of this augmentation technique. To the author’s knowledge, the application of the Mantel test and C2ST in this context has not been previously documented.

#### 2.4.1. Mantel Test

The Mantel test is a statistical technique employed to assess the correlation between distance matrices via the Mantel test [30]. It compares two distance matrices by calculating the correlation among the lower or upper triangles of the symmetric distance matrices. The terminology “distance matrix” is utilised in this research to refer to the correlation matrix between the features of a provided signal. The Mantel test makes it easier to investigate whether the produced signal can maintain the inherent relationship among features in the original signal. The present study employed Pearson’s product–moment correlation coefficient with 1000 permutations.

According to Mantel’s definition [31], the Mantel test calculates a test statistic $r_{M}$ based on two symmetric distance matrices, namely $D_{X}$ and $D_{Y}$. $r_{M}$ is defined as
(11)$$r_{M}=\frac{1}{d-1}\sum_{i=1}^{n-1}\;\sum_{j=i+1}^{n}\;\mathrm{stand}\;{\left(D_{X}\right)}_{ij}\mathrm{stand}\;{\left(D_{Y}\right)}_{ij}\;d=\frac{n(n-1)}{2}$$
where “n” denotes the total number of rows and columns in all the distance matrices. The matrices $\mathrm{stand}\;\left(D_{X}\right)$ and $\mathrm{stand}\;\left(D_{Y}\right)$ consist of standardised distances within their respective upper triangles and are both classified as distance matrices.

#### 2.4.2. Classification

One of the aspects determining the quality of the generated signal is its ability to help the classification model achieve a better classification result, which is one of the main reasons GANs and other signal synthesis models are used. This research used the LazyPredict library in Python, which contains 36 models divided into three categories: linear models, tree-based models and ensemble models. Five distinct sliding windows with varying increments were employed to extract features, and then, these features were used as inputs to the classifiers. The StandardScaler was employed for feature scaling and centring; it standardises the dataset by transforming it to have a mean of zero and a variance of one, resulting in enhanced model performance and facilitated convergence of the model [32]. The best results were obtained from ensemble models, in particular, the ADAboost and bagging classifier. AdaBoost, short for adaptive boosting, is a machine-learning algorithm, which combines weak classifiers to create a strong classifier by giving more importance to misclassified instances Bagging [33], short for bootstrap aggregating, is a machine-learning technique. Bagging classifiers use multiple independent classifiers trained on different subsets of data to improve prediction accuracy and reduce the impact of outliers or noisy data [34].

#### 2.4.3. Augmentation Classification Improvement

Synthetic signals generated from GAN are added incrementally to assess its effect on the classification results of the model used and to examine whether the augmentation can improve the classification result. The increments are 25% of the original signal. Each increment is balanced in terms of the amount of data for each gesture compared to the original data. Support vector machine (SVM) was used for classification in this section. SVM is an established technique for pattern detection in myoelectric signals. Its main purpose is to find an n-dimensional hyperplane, which divides a set of input feature points into distinct classes. With this method, complex patterns may be detected [35].

#### 2.4.4. Classifier Two-Sample Test

Two-sample tests are employed to ascertain the equality of two probability distributions and their similarity. C2ST can be utilised to accomplish the same objective, and a binary classifier can be employed to evaluate the quality of the produced distribution [36,37]. Osokin et al. [38] used classifier two samples to evaluate their GAN model, which was designed to produce synthetic medical images. Liu et al. [39] and Pacchiardi et al. [40] used C2ST for anomaly detection in videos.

Two samples are selected from distributions G and R, which represent the generated and reference distributions, respectively. The positive and negative labels are assigned to R and G. For the null hypothesis P = R to hold true, the results should remain close to the chance level, indicating that the two distributions are equivalent. The mean of the individual features determines the accuracy of the test and converges towards a Gaussian null distribution. Lopez-Paz et al. [36] designed a dataset comprising $x_{i},y_{j}\in X$, where i=1,…,n, and j=1,…,m. To determine the equality of G and R, the construction of the dataset is as follows:(12)$$D={\left\{\left(x_{i},0\right)\right\}}_{i=1}^{n}\cup {\left\{\left(y_{i},1\right)\right\}}_{i=1}^{m}≔{\left\{\left(z_{i},l_{i}\right)\right\}}_{i=1}^{n+m}\;$$

Dataset D is randomly shuffled and then divided into two distinct subsets, namely the training ($D_{tr}$) and testing datasets ($D_{te}$), which correspond to 80% and 20% of the dataset to avoid overfitting and obtain an accurate estimation of the classifier’s performance.

The training process of the binary classifier is f:X→[0,1] for $D_{tr}$, while $f\left(z_{i}\right)$ is the probability distribution, $p\left(l=1|z_{i}\right)$.
(13)$$\hat{t}=\frac{1}{n_{te}}\sum_{\left(z_{i},l_{i}\right)\in D_{te}}\;I\left[\left(f\left(z_{i}\right)>\frac{1}{2}\right)=l_{i}\right]\;$$

Throughout the GAN training process, the generator attempts to generate a probability distribution, which closely approximates that of the discriminator. Nevertheless, in actuality, the GAN will not generate an identical distribution or serve as a mere replica of the input. Lopez-Paz et al. [36] suggested using a margin classifier with a finite norm. Consequently, in the present investigation, the margin classifier with a finite norm implemented was logistic regression due to its ease of training and utilisation.

## 3. Results

Visually, the GAN attempts to produce signals in Figure 7 mimicking the original signals. However, a complete duplicate of the original signal must be avoided, as this would signify mode collapse. Mode collapse is a bad behaviour, which shows a lack of diversity in the samples, information loss and an inability to cover the complete target distribution. As a result, the quality of the generated signals will not be assessed based on the original sEMG signals. Instead, features from the time and time-frequency domains will be retrieved and analysed. The relationships between features in the reference and generated signals are examined in Section 3.1. We evaluate the categorisation precision of both sets of signals in Section 3.2. The improvement in classification accuracy attained through the augmentation of the generated signal is covered in Section 3.3. Last but not least, Section 3.4 compares the distributional similarity between the reference and generated signals using a classifier two-sample test. By using this structured technique, the generated signals can be thoroughly evaluated, along with their impact on classification accuracy and their statistical resemblance to the reference signals.

### 3.1. Mantel Test Analysis

This visual method of examining the correlation matrix is laborious and impossible for a large amount of data. Therefore, the Mantel test is utilised, which describes the similarity between the two matrices using a correlation coefficient and a *p*-value. The *p*-value of the Mantel test is the probability of obtaining a correlation coefficient as extreme as the one observed, assuming that the null hypothesis of no correlation is true. A low *p*-value (typically < 0.05) indicates that the correlation is statistically significant.

Figure 8 represents the correlation coefficient between correlation matrices of the original and synthetic signals for both datasets and the corresponding *p*-value for both the public and private datasets. For both datasets, the vast majority of the correlation coefficient indicates a positive correlation with a *p*-value < 0.05 for the public dataset, indicating that the results are statistically significant. In terms of the private dataset, a minimal number of samples have a *p*-value > 0.05, indicating that very few of the results are statistically insignificant.

For the private dataset, 9% of the samples showed a very weak correlation; 31% of its samples showed a weak positive correlation; 36% showed a medium correlation; 23% showed a strong correlation; and 1% showed a very strong correlation. In terms of the public dataset, 20% of the samples showed a very strong correlation; 53% showed a strong correlation; 23% showed a medium correlation; and the rest were in a weak correlation range with only 4%.

Both the private and public datasets show a positive correlation. All *p*-values for the public dataset are less than 0.05; for the private dataset, 96.3% have values less than 0.05. This indicates that the null hypothesis is rejected for most of the samples in both datasets, and the results are statistically significant. Hence, it can be deduced that the GAN model used has the capability of producing samples, which exhibit a resemblance of the complex interrelationships among the diverse features of the original signal.

### 3.2. Classification

In order to further analyse the synthetic and real data, both synthetic and reference data were classified on their own using the LazyPredict library. The models were trained on 80% of the dataset to avoid overfitting and obtain an accurate assessment of their performance. The remaining 20% of the dataset was used to evaluate the accuracy of the models, as illustrated in Figure 9. The best results were achieved in ensemble models AdaBoost and bagging classifier.

The classification models were trained to predict the correct gestures per subject, using the features extracted from each window for that subject. The windows were applied in all trials and only in the gesture part of the trial and not the rest part. The gestures were 3 s long and repeated 15 times. The classification results of each subject for all windows were then averaged across both datasets and presented in Figure 9.

The generated sample of the private dataset achieved its highest classification accuracy of 98.4% using a window size of 1000 and an increment of 250 data points versus 95.6% for the original signal for the same window and increment with a difference of 2.8%. Overall, the average difference between the generated and reference samples was 6.9%.

In terms of the public dataset, the highest classification accuracy achieved was 89.6% for the sample generated with a window size of 1000 and an increment of 250 data points as well versus 88.4% accuracy for the original signal of the same window and increment with a difference of 1.2%. The average difference among all classified samples was 8.9%.

For both datasets (Figure 9a,b), the higher classification results for the synthetic signals can be attributed to the fact that the synthetic data generated from GAN are more informative and easier to classify than the original data because the intraclass distance is small and the interclass distance is large compared to that of the original signal. A small intraclass distance makes it easier for the classifier to find patterns within each class, and a substantial interclass distance makes it easier for the classifier to distinguish between different classes. This indicates that the GANs are able to learn the underlying distribution of the data and generate samples, which are more representative of the real world, and that the machine-learning model is able to generalise to the target distribution.

### 3.3. Augmentation Classification Performance Analysis

Signals were divided into five different window sizes with lengths of 200, 400, 600, 800 and 1000 data points, which corresponded to 50, 100, 150, 200 and 250 milliseconds, respectively, for the public dataset and 100, 200, 300, 400, 500 for the private dataset, as seen in Table 1. The window increments used were 25%, 50%, 75% and 100% of the specific window size. The features from these windows were extracted and used in this classification. The datasets were divided into 80% for training and 20% for testing to obtain an accurate assessment of the model performance; the accuracy results recorded were those of the testing dataset. The testing dataset comprised a balanced distribution of both generated and original signals. Specifically, the percentage of generated signals in the testing dataset was consistent with the percentage of generated signals present in the entire mixed dataset. Windows were applied in all trials exclusively for the gesture segment, excluding the rest portion. The length of a gesture was 3 s, as mentioned in Figure 2, and it was repeated 15 times.

In order to study the effect of the generated synthetic signals on the performance of a classifier, 25%, 50%, 75% and 100% of the synthetic signals are added to the classification dataset, and the classification results are plotted in Figure 10 below.

In Figure 10, it can be observed that the window increment of 25% of all windows achieved the highest results in both datasets. The window with the highest classification result was the one with a length of 1000 data points and an increment of 250 data points for both public and private datasets, with a classification accuracy of 98.5% and 98.1% using the SVM classifier, respectively. The best classification result corresponded—for both datasets—to 100% addition of synthetic data. The addition of generated samples to the original dataset improved the classification results by an average of 1.21% and 5% for the private and public datasets, respectively.

### 3.4. Classifier Two-Sample Test

Logistic regression is used for classification to investigate the similarity of two probability distributions, and it is widely used and easy to train. The results for both datasets are presented in Figure 11a,b. The private dataset’s average accuracy for each window is presented in Figure 11a; the average accuracy of all windows is 59%. In terms of the public dataset, the average accuracy can be found in Figure 11b, and the average of all window accuracies is 66%.

While the average in both datasets is higher than the threshold of 50%, it is still close to it, especially for the private dataset. This indicates that while the generated data are not perfectly similar to those of the reference signal, they are similar to some extent.

## 4. Discussion

To the best of the author’s knowledge, the use of 1DDCGAN for the generation of sEMG signals corresponding to hand gestures has not been previously explored. Additionally, two novel metrics were introduced—namely the Mantel test and C2ST—for evaluating the quality of the generated signals. These metrics have not been utilised in this context before.

The majority of the Mantel test results indicated a medium-to-strong correlation with a *p*-value less than 0.05, indicating that the correlation between two matrices is statistically significant at a 95% confidence level, meaning that there is strong evidence to reject the null hypothesis of no correlation between the matrices. The Mantel test results suggest that the proposed GAN successfully captured the inherent relationship between the extracted features in the real signal and replicated it in the generated signal. This finding demonstrates the effectiveness of the GAN model used in accurately reflecting the underlying patterns of the real data and in generating high-quality signals.

In Section 2.3, the synthetic and original datasets were independently subjected to classification using multiple classifiers. The classification results were found to be the best when using a data window size of 1000 and an increment of 250. In this configuration, the generated data achieved a classification accuracy of 98.4%, while the original data achieved a lower accuracy of 95.6%. Consequently, the generated data outperformed the original data by about 2.8% in classification. The results indicate that the generated data consistently exhibited higher accuracy compared to the original data across all signal classifications. This suggests that the proposed 1DDCGAN model effectively produced signals characterised by small intraclass distance and large interclass distance, thereby facilitating easier classification. The observed outcome is consistent with the expected output of a GAN, suggesting its ability to generate signals, which show similarity within a given class while also demonstrating sufficient dissimilarity from other classes, resulting in notable classification performance.

Regarding the augmentation test in Section 3.3, the synthesised data were augmented by adding increments of 25% of the original signal’s length until reaching a ratio of 1:1 with the original signal. Subsequently, the augmented data were classified using an SVM classifier. The results indicate that the augmentation technique led to enhanced classification outcomes for all of the windows. It is noteworthy to mention that the best outcomes were also attained when using a window size of 1000, with increments of 250 data points, which aligns with the findings of the classification analysis discussed in Section 3.2. The overall improvement varied between 1.21% and 5% for both the public and private datasets, with a maximum improvement of 12.2%.

In terms of the C2ST results, the perfect result would be as near to the chance level as possible, as this would indicate that the two samples are indistinguishable. The results obtained for the private dataset and the public dataset were 58% and 66%, respectively. These results suggest that there is a sufficient degree of similarity between the generated signals and the original signals, indicating that the proposed model is capable of producing signals, which closely resemble the original data. The classification results show that the augmentation of the generated dataset can enhance the classification result.

In summary, the evaluation metrics results demonstrated that the proposed model outperformed the real data in classification while effectively reflecting the underlying patterns of the real data. The results of the classification were improved once the generated dataset was augmented, showing that GAN-generated samples can offer more varied and representative examples for the classification model to learn from.

## 5. Conclusions

The approach put forth in this study involved using 1DDCGAN to generate signals. In addition, multiple evaluation metrics were used to assess the generated data. The metrics used were the Mantel test, classification, augmentation test and C2ST. The Mantel test and C2ST are novel in this context. The results indicated that the produced signals were able to capture the complex relationship among features and showed a smaller interclass distance and a large intraclass distance. The ability of 1DDCGAN to produce signals varied, and representative examples were shown in the augmentation test, where the augmentation of the synthetic signals achieved an enhancement in the classification results of about 1.21–5%. Furthermore, the C2ST results showed the ability of 1DDCGAN to produce signals similar to the original ones.

The findings indicate that the utilisation of 1DDCGAN demonstrated efficacy in generating signals suitable for augmenting signals in AI-controlled prostheses. This augmentation process has the potential to reduce the duration of laboratory training required for amputees to train the classification model of the prosthesis. Consequently, this approach offers benefits, such as increased comfort and time savings for amputees, as well as improved classification outcomes. Furthermore, the 1DDCGAN technique can also be employed to supplement datasets, which contain missing or corrupted data. Additionally, the Mantel test outcomes revealed that the generated signals exhibited a comparable interrelationship among their features compared to the original signals. This finding is significant, as it indicates that the 1DDCGAN model has the ability to generate signals with similar characteristics, making it a dependable tool for analysis and research purposes. Moreover, the model ensures the privacy of signal owners, thereby enabling researchers to access valuable data, which can contribute to advancements in the field of sEMG signal classification for gesture recognition.

## Figures and Tables

**Figure 1 bioengineering-10-01353-f001:**
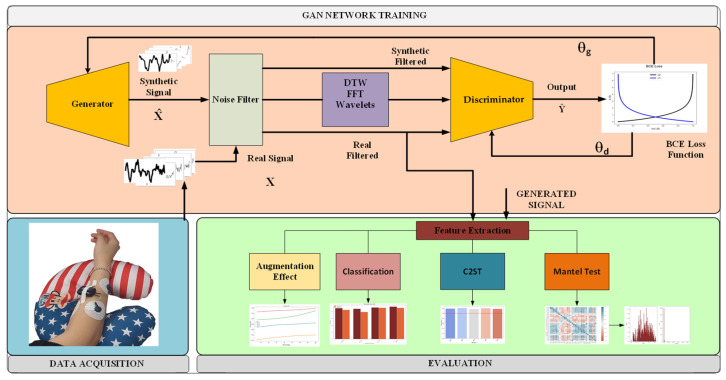
Flow chart of the study. The data acquisition stage (lower left) is followed by a GAN training stage in the top section. The evaluation stage (lower right) evaluates the data quality using different techniques.

**Figure 2 bioengineering-10-01353-f002:**
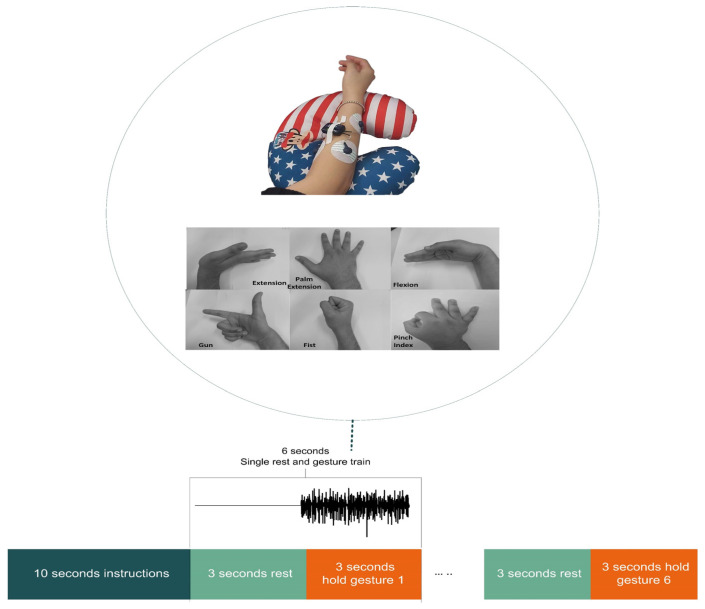
Electrodes’ placement during an acquisition session and the six gestures for the private dataset. The timeline of a typical session is 3 s for rest and 3 s for gestures, each repeated 15 times.

**Figure 3 bioengineering-10-01353-f003:**
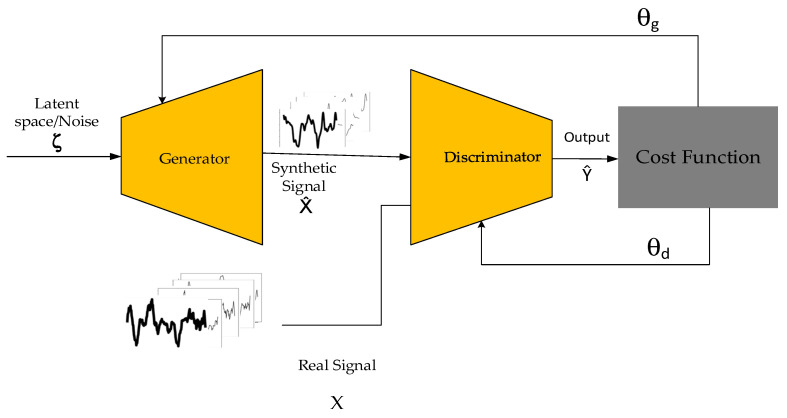
GAN components.

**Figure 4 bioengineering-10-01353-f004:**
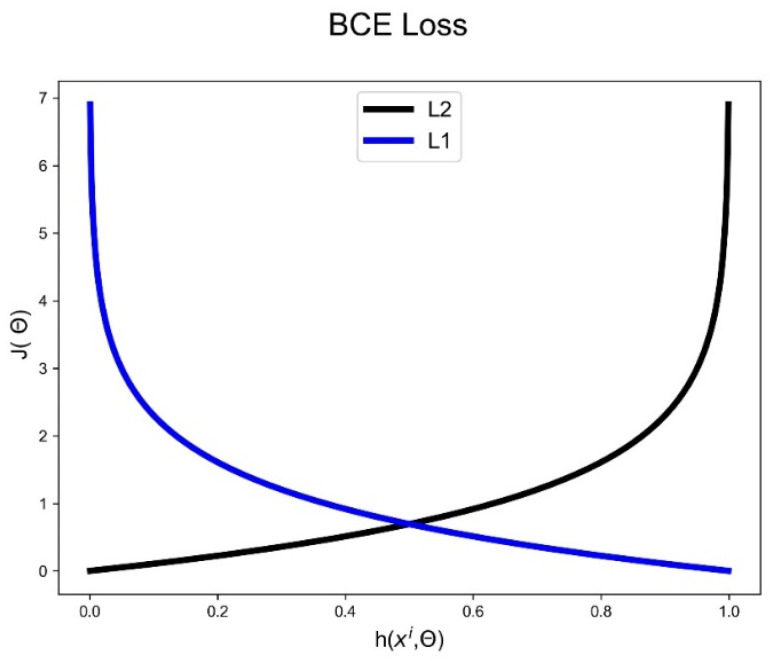
The graph of the two terms comprising the BCE loss function.

**Figure 5 bioengineering-10-01353-f005:**
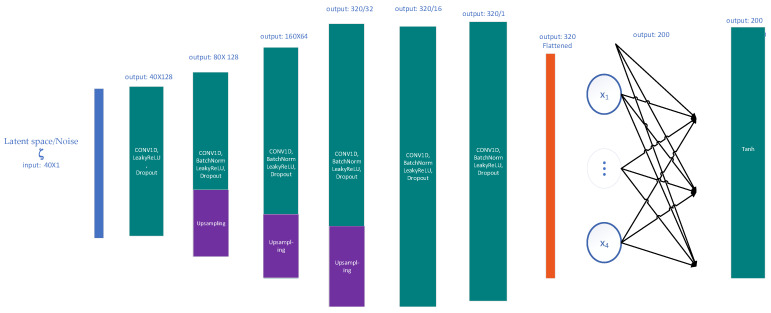
Generator architecture.

**Figure 6 bioengineering-10-01353-f006:**
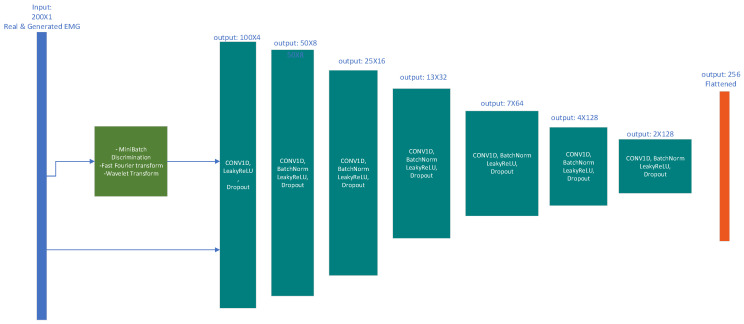
Discriminator architecture.

**Figure 7 bioengineering-10-01353-f007:**
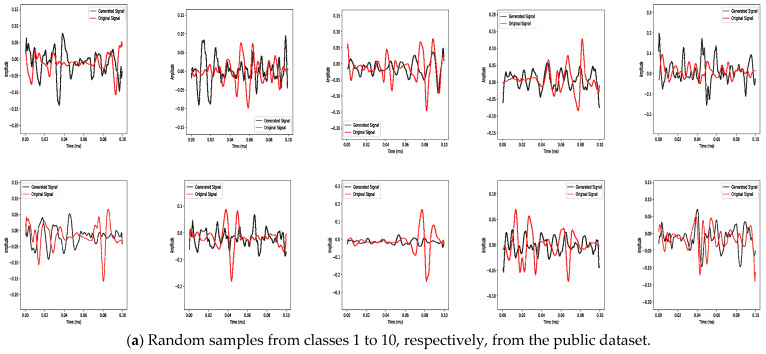
Random samples of reference signals and their corresponding generated signals of all classes of both datasets.

**Figure 8 bioengineering-10-01353-f008:**
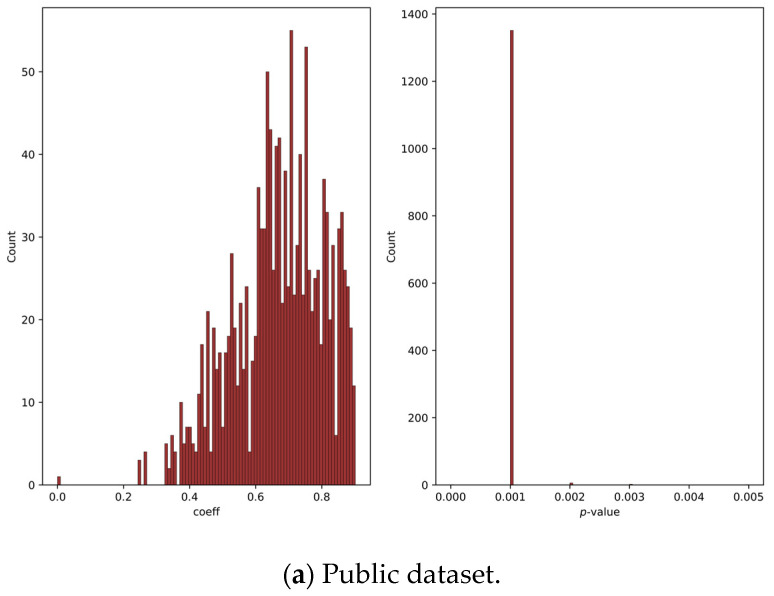
Mantel’s correlation coefficient and *p*-value for public and private datasets. The horizontal axis denotes the numerical value of the *p*-value, while the vertical axis indicates the frequency of its occurrence.

**Figure 9 bioengineering-10-01353-f009:**
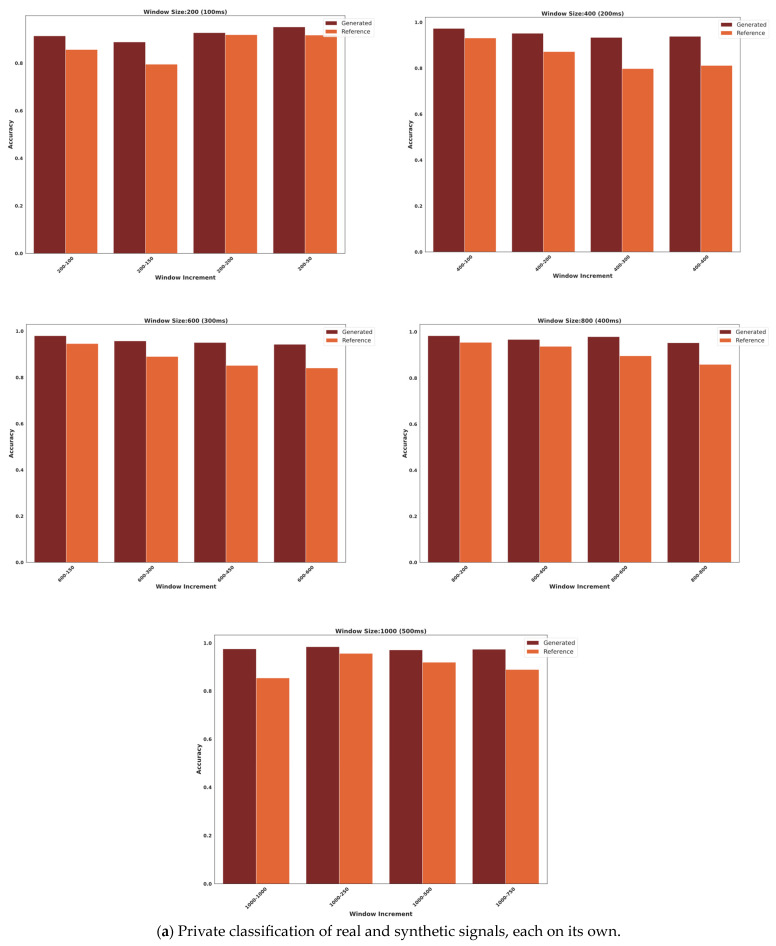
Accuracy results of private dataset (**a**) and public dataset (**b**).

**Figure 10 bioengineering-10-01353-f010:**
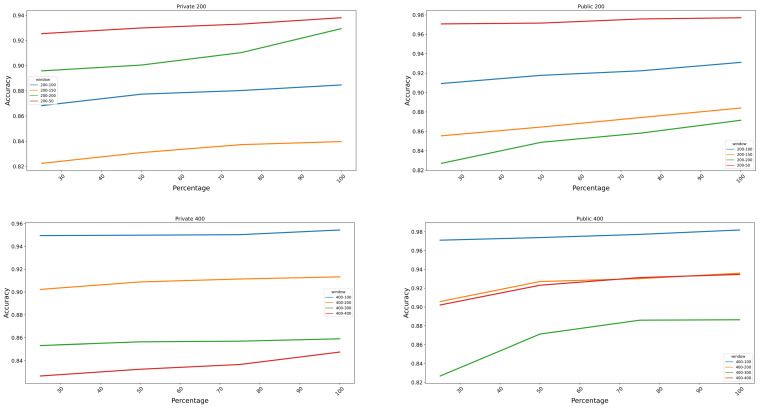
The figure represents the classification results of five different windows with four increments representing 25%, 50%, 75% and 100% of each window for both public and private datasets. The X-axis represents the percentage of synthetic signals added to the original signal during classification.

**Figure 11 bioengineering-10-01353-f011:**
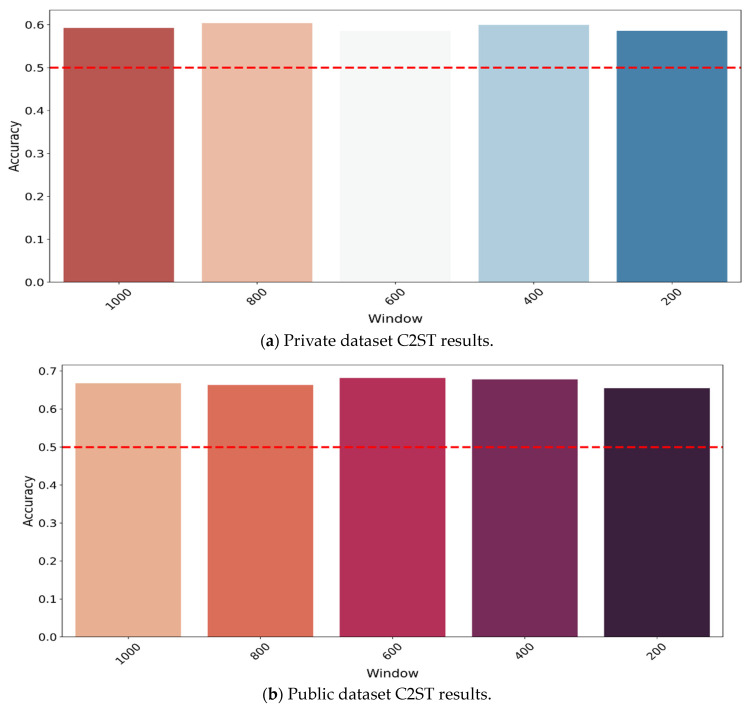
Private (**a**) and public (**b**) dataset C2ST results.

**Table 1 bioengineering-10-01353-t001:** The number of data points for both public and private datasets and the corresponding lengths of these data points. The datasets have different frequencies: 4000 Hz for the public and 2000 Hz for the private dataset.

Number of Data Points	Public Dataset (ms)	Private Dataset (ms)
200	50	100
400	100	200
600	150	300
800	200	400
1000	250	500

## Data Availability

The data presented in this study will be made available upon request to the authors.

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
