# Peer review of "Synthesis of sEMG Signals for Hand Gestures Using a 1DDCGAN"

_bioengineering, 2023, doi:10.3390/bioengineering10121353_

Round 1

Reviewer 1 Report

Comments and Suggestions for Authors

1. The author should elaborate more on the training/test set partition for classification. Specifically, whether the generated signal data was used for testing.

2. The table 2 comparison with other methods is not valid from my perspective. The methods' improvement on different datasets cannot validate the supremacy of the proposed method. I would recommend implement/reproduce those methods on authors' dataset.

3. in 3.1 the authors claim the best results were obtained from ensemble methods but in table 2 the classification method is noted as SVM. The discrepancy should be elaborated.

Comments on the Quality of English Language

The presentation/grammar of the current manuscript is satisfactory.

Reviewer 2 Report

Comments and Suggestions for Authors

The aim of the paper is focused on the evaluation of GAN performance in synthetizing EMG data. The aspects raised by authors are of interest for research and for some specific application, as for instance myoelectric control of exoskeleton and limb prosthesis due to the difficulties to collect data from pathological groups. Furthermore, the experimental procedure for pathological subjects can lead to fatigue and stress.

Despite these issues, the paper is not easily readable, and the authors report the results presented in literature unclearly. The novelty of the present study is not understandable. Several methods, to generate data and to validate the model are cited and applied without providing the grounds for the methodological choices.

General comments

1.       Ninapro dataset considered for this study (and in general in all Ninapro datasets), contains data captured from surface electrodes and in this case data are recorded by control and pathological subject (amputees). The relevance of this kind of dataset is not related to the EMG technology but to the fact that there are a large amount of data recorded with pathological subjects and this aspect it is not easily obtainable.

2.       Moreover the surface EMG technology provides a global information on the muscle activity while the intramuscular electrodes, thanks to their natural selectivity, gives the single muscle fiber activity.

3.       Privacy issues related to the signal recording, processing and storing have to be treated following experimental good practices and cannot limit the use of data. The data have to be anonymized both if collected from a control group and from a pathological group and in general this aspect does not limit data use if correct practice foe their collection are followed.

4.       The authors reported results of studies in a confusing manner. The use of GAN for the bio-signal generation is cited together with the use for image generation and processing and the aim is confused.

5.       Authors should clarify how the techniques cited in lines 105-107 can be use to evaluate generated signals.

6.       What is the rationale for augmentation effect, classification, C2ST and manual test steps? In materials and methods section after the flow-chat they are mentioned but any explanation about their use is provided.

7.       The electrodes position as reported in figure 2 seems to indicate a very large inter-electrodes distance. The guidelines for sEMG signal recording (see for instance SENIAM documentation) usually suggest to reduce such distance. Can the authors explain the rationale for this choice?

8.       Given that, the visual analysis of correlation matrix is not recommended when large amount of data has to be processed, authors should avoid to present results related to it.

9.       The authors should better clarify what they mean when they show accuracies for different window sizes. If they refer to the dimension of the windows for features calculation, they should specify if the repetitions of each task for one experimental trial were segmented and if rest is analyzed separately from the motion portion or if the whole trial has been examined without isolating the rest from the movement. If this is the case they also should specify the length of the trial and the number of trials used for each window.

Minor Comments

The flow-chart of the study showed in figure 1, is not clear and does not provide an evident flow of the steps.

Figure 7 show a very short portion of the signals (real and generated) for 0.1ms. Moreover it is not clear why the authors decided to report 16 diagrams where signals (generated and real) if then they state (line 432) that “ the complete duplicate between original and generated signal has to be avoided”. Any relation with the class is missing and it appears that a single figure was sufficient to support their sentence.

In figure 10, authors reported accuracy but it is not clearly state if this values are related to the training or testing phase.

Comments on the Quality of English Language

no specific comments

Reviewer 3 Report

Comments and Suggestions for Authors

  • The Introduction section needs to be more extensive. It should be divided so that the study of the state of the art is presented in a separate section that refers to the background. This would make it easier to read and understand the content.

  • In addition, it is suggested to include a description of the manuscript's structure at the end of the Introduction section.

  • Improve the quality of Figure 2.

  • Subsection 2.2 appears to be empty. Please check the numbering of the sections. Also, subsection 3.2 and 2.4.2 has the same name.

  • Figures must be vectorized and some are in low quality (for example, Figure 7).

  • Lines 515-518: I suggest to justify dividing the dataset into 80% for training and 20% for testing when making the classification model. This ensures the choice is appropriate for the data set and the problem we are addressing.

  • Lines 31-32 can be justified with the following works related with EMG signals: A study of computing zero crossing methods and an improved proposal for emg signals; A novel methodology for classifying emg movements based on svm and genetic algorithms; Optimizing emg classification through metaheuristic algorithms; Support vector machine-based emg signal classification techniques: a review a study of movement classification of the lower limb based on up to 4-emg channels.

  • Regarding classification models, mention is made of the LazyPredict library, which includes various models such as Random Forest or SVM. However, detailed explanations of these models should be provided in the materials and methods section to give the reader a complete understanding of the work.

Round 2

Reviewer 2 Report

Comments and Suggestions for Authors

The authors addressed  my previous comments. I agree with their response. I remain not completely satisfied with respect to EMG sensors position. The authors placed bipolar sensors in some arm muscles. The distance between sensors should be minimised to reduce input impedance as also stated in litterature (SEINAM Guiddlines). The answer of the authors about this point it is not centered.

Comments on the Quality of English Language

No specific comments about the quality of English language
